# Tracing the oxygen isotope composition of the upper Earth's atmosphere using cosmic spherules

Andreas Pack[1], Andres Höweling[1,2], Dominik C. Hezel[3], Maren T. Stefanak[1], Anne-Katrin Beck[1],
Stefan T.M. Peters[1], Sukanya Sengupta[1], Daniel Herwartz[3] & Luigi Folco[4]

Molten I-type cosmic spherules formed by heating, oxidation and melting of extraterrestrial Fe,Ni metal alloys. The entire oxygen in these spherules sources from the atmosphere. Therefore, I-type cosmic spherules are suitable tracers for the isotopic composition of the upper atmosphere at altitudes between 80 and 115 km. Here we present data on I-type cosmic spherules collected in Antarctica. Their composition is compared with the composition of tropospheric $O_2$. Our data suggest that the Earth's atmospheric $O_2$ is isotopically homogenous up to the thermosphere. This makes fossil I-type micrometeorites ideal proxies for ancient atmospheric $CO_2$ levels.

[1] Universität Göttingen, Geowissenschaftliches Zentrum, Goldschmidtstraße 1, 37077 Göttingen, Germany. [2] Karlsruher Institut für Technologie, Institut für Angewandte Materialien - Werkstoffprozesstechnik, Hermann-von-Helmholtz-Platz 1, 76344 Eggenstein-Leopoldshafen, Germany. [3] Universität Köln, Institut für Geologie und Mineralogie, Greinstraße 4-6, 50939 Köln, Germany. [4] Universitá di Pisa, Dipartimento di Scienze della Terra, Via Santa Maria 53, 56126 Pisa, Italy. Correspondence and requests for materials should be addressed to A.P. (email: apack@uni-goettingen.de).

Free molecular oxygen ($O_2$) is released by photosynthesis into the atmosphere and is essential for all breathing animals. With exception of data for the last 800,000 years from air inclusions in polar ice, little direct information is available about concentration and isotope composition of ancient atmospheric $O_2$. This is due to the limited interaction between air molecular oxygen and the lithosphere.

Among the rare rocky materials that contain atmospheric oxygen[1–3] there are particular types of micrometeorites (microscopic extraterrestrial dust particles) called cosmic spherules[4]. Roughly 10 tons of small extraterrestrial particles are deposited onto the Earth's surface per day[5]. The particles collide with the Earth's atmosphere at velocities of 11–70 km s$^{-1}$ (ref. 6) and are visible as shooting stars when they are decelerated and at altitudes up to ~80–115 km[7,8]. A portion of these extraterrestrial particles totally melts during the atmospheric entry and is termed cosmic spherules. Cosmic spherules that are composed of Fe,Ni oxides are termed 'I-type cosmic spherules' (in the following, we use the short version 'I-type spherules'[9–11]. These I-type spherules formed by oxidation of extraterrestrial Fe,Ni metal alloys, which are ubiquitous components of meteorites.

Because oxygen in I-type spherules originates entirely from the atmosphere, they are excellent probes for the isotopic composition of upper atmospheric oxygen. The isotopic composition of atmospheric oxygen, in turn, is a proxy for the global primary production (GPP) and atmospheric $CO_2$ levels[1,2,12–14]. It is not clear, however, if the atmospheric oxygen is isotopically homogenous up to the meso- and thermosphere, where cosmic spherules interact with air.

The stable isotope composition of tropospheric $O_2$ (99.8% $^{16}O$, 0.04% $^{17}O$, 0.2% $^{18}O$) is controlled by the steady state between photosynthesis and respiration (mass-dependent Dole effect[15]), evapotranspiration and mass-independent fractionation in the stratosphere[14,16]. For the composition of the modern troposphere values of $23.4 \leq \delta^{18}O \leq 24.2$‰ and $-0.566 \leq \Delta'^{17}O \leq -0.430$‰ have been reported in the literature[14,17–20] (for definitions, see Methods) with little variations up to 61 km (ref. 17). The high $\delta^{18}O$ of tropospheric $O_2$ is caused by the Dole effect, whereas the low $\Delta'^{17}O$ value reflects mass-independent fractionation effects in the stratosphere. The higher the atmospheric $CO_2$ levels, the lower the $\Delta'^{17}O$ values[12,14]; a relation that was used as paleo-$CO_2$ barometer[1,2,13].

No experimental oxygen isotope data are available for the upper atmosphere at altitudes >61 km. To obtain information about the $\Delta'^{17}O$ heterogeneity of the atmosphere, we measured the oxygen ($\delta^{17}O$, $\delta^{18}O$) and iron ($\delta^{56}Fe$, $\delta^{57}Fe$; see Methods for definition) isotope composition of I-type spherules from the Transantarctic Mountains that have ages <2 Ma (ref. 21). For this time interval, atmospheric $CO_2$ levels as well as $\Delta'^{17}O$ of $O_2$ did not deviate much from the modern level[22,23] and data can be used to test whether the atmosphere is isotopically homogenous. Because of the ~1,000 years residence time of atmospheric $O_2$ (ref. 14), no effect on the man-made increase in $CO_2$ is yet visible in decreasing $\Delta'^{17}O$.

The oxygen isotope composition of I-type spherules is controlled by the composition of the oxidizing species (for example, atmospheric $O_2$), the fractionation during oxidation of the Fe,Ni alloys, and the fractionation during atmospheric evaporation.

Cosmic spherules have higher $\delta^{18}O$ values (up to 56‰; ref. 24) than any terrestrial material reported so far. The high $\delta^{18}O$ led Clayton et al.[4] to propose a heavy oxygen isotope reservoir in the upper atmosphere. Davis et al.[25], however, showed that I-type spherules are also enriched in heavy iron isotopes with extreme $\delta^{56}Fe$ values of up to 45‰. They concluded that high $\delta^{18}O$ values

are due to evaporation and do not reflect the isotope composition of the upper atmosphere; a conclusion that was supported by further measurements[24,26–29]. From the oxygen and iron isotope composition, Engrand et al.[24] modeled evaporative mass losses for I-type spherules of 54–85%.

Because evaporative fractionation is strictly mass-dependent, I-type spherules still provide unique information about the mass-independent anomaly in $\Delta'^{17}O$ of their upper mesospheric oxygen source. The reconstruction of variations in atmospheric $\Delta'^{17}O$ from fossil cosmic spherules[4,30–33] would be a new paleo-$CO_2$ proxy. The only published $\Delta'^{17}O$ data on I-type spherules by Clayton et al.[4] and Engrand et al.[24], however, have intrinsic uncertainties that are too large (0.1 to >1‰) to provide reliable information on the composition of the upper atmosphere.

We present new high-precision oxygen isotope data of tropospheric $O_2$ and compare these data with new high-precision oxygen and iron isotope data from Antarctic I-type spherules. These data are combined with results of oxidation and evaporation experiments to test if the Earth atmosphere is isotopically homogenous and if isotope ratios of fossil cosmic spherules are suitable paleo-$CO_2$ proxies.

## Results

**Oxygen isotope composition of tropospheric air.** The mean composition of air oxygen from our study (series B; Supplementary Table 1) is $\delta^{18}O = 24.15 \pm 0.05$‰ with $\Delta'^{17}O = -0.469 \pm 0.007$‰. An earlier protocol (series A; Supplementary Table 1) gave identical $\Delta'^{17}O$, but slightly lower $\delta^{18}O$. The datum from this study is within the range reported in the literature and agrees with $\Delta'^{17}O$ values of Thiemens et al.[17] and Young et al.[14]. The measured data of Young et al.[14] have been corrected relative to the San Carlos olivine value reported by Pack et al.[34]. The corrected measured datum of $\Delta'^{17}O = -0.467 \pm 0.005$‰ for air oxygen[14] is then identical to our measured datum of $-0.469 \pm 0.007$‰ (Supplementary Table 1) and agrees with the model datum of $-0.469$‰ presented by Young et al.[14]. The non-application of VSMOW2-SLAP2 scaling[34] to our air data shifts the $\Delta'^{17}O$ down to ~$-0.50$‰, which would be closer to the values of Barkan and Luz[18] and Kaiser and Abe[20]. For this study, we adopt $\delta^{18}O = 24.15$‰ and $\Delta'^{17}O = -0.47$‰ for the troposphere.

**Oxygen and iron isotope composition of cosmic spherules.** The isotope composition of the I-type spherules and the oxidation experiment run products are listed in Supplementary Table 2. The $\delta^{18}O$ of the spherules ranges from 36 to 42‰. The corresponding $\Delta'^{17}O$ ranges from $-0.72$ to $-0.62$‰. The $\delta^{18}O$ and the $\Delta'^{17}O$ is within the range reported by Clayton et al.[4] and Engrand et al.[24]. The $\delta^{56}Fe$ values are high for all spherules, ranging from 22 to 32‰ (Supplementary Table 2).

## Discussion

The interaction between cosmic Fe,Ni metal and the Earth atmosphere during deceleration is considered to proceed in two consecutive steps. The first step is the atmospheric heating and oxidation of the infalling Fe,Ni metal alloy (fractionation in oxygen isotopes only). The second step is the melting and evaporation of the Fe,Ni oxides (fractionation in both, oxygen and iron isotopes).

Information about the oxygen isotope fractionation that is associated with the oxidation step is obtained from experiments (this study; see Methods) and from iron meteorite fusion crust data[4,35,36]. The products of the high-$T$ metal oxidation experiments have $\delta^{18}O$ values that are 4‰ lower than air oxygen (Supplementary Table 2). Clayton et al.[4] reported $\delta^{18}O$

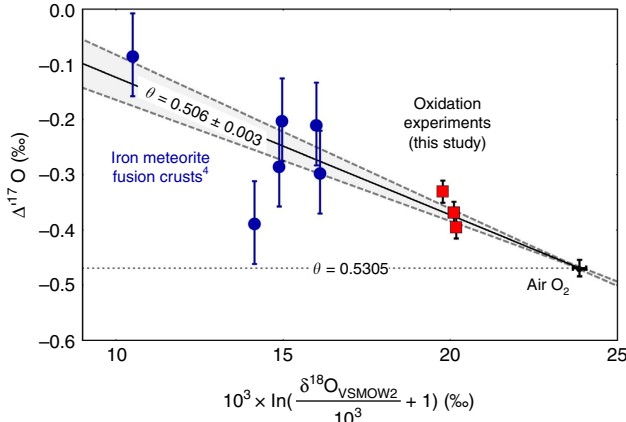

**Figure 1 | Triple oxygen isotope fractionation during metal oxidation.** Plot of $\Delta'^{17}O$ versus $\delta^{18}O$ of the run products of the high-$T$ oxidation experiments (solid red squares) along with air oxygen (this study) and iron meteorite oxide fusion crust (solid blue circles;[4]; the lowest values were not considered as it was likely affected by exchange with low-$\delta^{18}O$ water). Shown are values with $1\sigma$ error bars. The oxidation is associated with a kinetic isotope fractionation with slope $\theta_O = 0.506 \pm 0.003$ (solid line; dashed lines indicate the $1\sigma$ s.d. uncertainty interval). The high-$T$ equilibrium slope $\theta = 0.5305$ (dotted horizontal line) is shown for comparison.

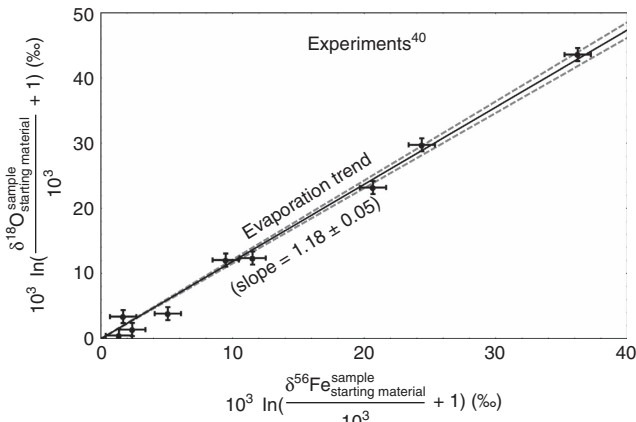

**Figure 2 | Oxygen and iron isotope fractionation during evaporation.** Plot of the linearized $\delta^{18}O$ versus $\delta^{56}Fe$ of evaporation experiments by Wang et al.[40]. The iron isotope fractionation is a function of fraction $f$ evaporated (Equation 1). The close coupling between $\delta^{18}O$ and $\delta^{56}O$ allows prediction of the pre-evaporation $\delta^{18}O$ of the I-type spherules. The $1\sigma$-errors (s.d.) were estimated on base of the scatter of the data. The regression line (solid) is shown with its respective $1\sigma$-error envelope (dashed lines).

values of iron meteorite fusion crusts that were $\sim9$‰ lower than air oxygen. They concluded that the atmospheric oxidation of Fe,Ni alloys is associated with kinetic fractionation and formation of isotopically light oxides with $\delta^{18}O_{Fe,Ni\ oxide} < \delta^{18}O_{air}$, which we also suggest as the cause of the low $\delta^{18}O$ of the experimental Fe,Ni oxides. Kinetic fractionation should be associated with a low $\theta_O$ value (for definition, see Methods). The best-fit line through air, experimental run products and iron meteorite fusion crust[4] has a slope of $\theta_O = 0.506 \pm 0.003$ (Fig. 1). This low $\theta$ value clearly supports that kinetic fractionation is the cause for the difference in $\delta^{18}O_{Fe,Ni\ oxides}$ and $\delta^{18}O_{air\ O2}$ (ref. 37). Molecular diffusion of $O_2$ would give a slope of $\theta_O = 0.508$. Iron meteorite fusion crust data[35] suggest that iron isotopes do not fractionate during oxidation.

For the experiments and iron meteorite fusion crusts, atmospheric oxygen is the oxidant. Above the ozone layer, however, a considerable fraction of molecular oxygen is steadily dissociated into atomic oxygen (for example,[38]). Atomic oxygen is a hazard for low Earth orbit space flights due to its highly corrosive nature. Because I-type spherules are oxidized at high altitudes, atomic oxygen may have contributed to the oxidation. However, Clayton et al.[4] and Genge[10] stated that no discrimination between atomic and molecular oxygen is likely during oxidation upon atmospheric entry because the collision energy between infalling meteoroids and air particles is higher than the $O_2$ bond strength. This implies that I-type spherules sample the bulk upper atmosphere oxygen (O and $O_2$).

Our experiments and the fusion crust literature data[4] show that $\alpha_{oxidation} < 1$ (for $^{18}O/^{16}O$; see Equation 5), but also reveal considerable variation. For I-type spherules, we assume that $0.9428 \leq \alpha_{oxidation} \leq 1$. The lower limit is given by pure Graham's law[39] fractionation with atomic oxygen being the moving species.

The second process affecting the isotopic composition of I-type spherules is evaporation[25]. Because iron isotopes are not affected by oxidation[35] but only by evaporation, $\delta^{56}Fe$ can be used as monitor for the degree of evaporation $f$ (refs 24,25,40; Equation 1). The $\delta^{56}Fe$ of the infalling metal is assumed to be

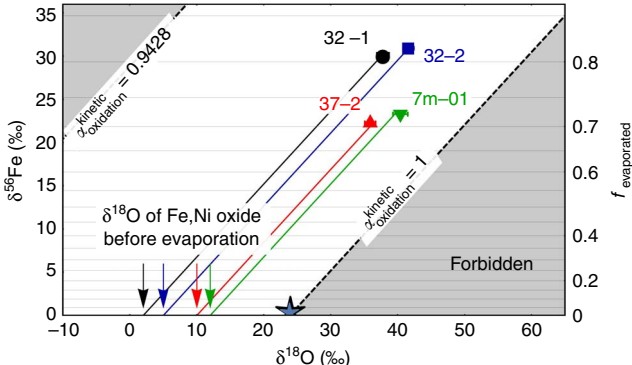

**Figure 3 | Model for the triple oxygen isotope fractionation during oxidation.** Plot of measured $\delta^{56}Fe$ versus $\delta^{18}O$ of spherules for the reconstruction of the pre-evaporative $\delta^{18}O$ (32-01: black solid circle; 32-02: blue solid square; 37-02: red solid triangle; 7 m-01: rotated green solid triangle). The right axis displays the fraction $f$ that evaporated (70–82%). The slopes of the lines (32-01: black; 32-02: blue; 37-02: red; 7 m-01: green) displayed were taken from the experiments by Wang et al.[40]) (Fig. 2. The grey shaded 'forbidden' areas outline cases for unreasonable kinetic fractionation factors $\alpha$ associated with atmospheric oxidation of Fe,Ni metal; all studied spherules fall outside the forbidden areas. The star marks the $\delta^{18}O$ of modern air oxygen.

$0 \pm 1$‰ relative to the IRMM-014 standard material[41].

$$\delta^{56}Fe = 10^3(1-f)^{\alpha_{evaporation}^{56/54} - 1} - 1 \qquad (1)$$

Wang et al.[40] determined an $\alpha_{evaporation} = 0.9820$ ($^{56}Fe/^{54}Fe$). This value in combination with $\delta^{56}Fe$ allows calculation of $f$ for the I-type spherules (Equation 1). The $\delta^{56}Fe$ values of the spherules from this study (Supplementary Table 2) indicate evaporative mass loss of $70 \leq f \leq 82$%, which is well in the range of $54 \leq f \leq 85$% reported by Engrand et al.[24].

The experimental relation between $\delta^{18}O$ and $\delta^{56}Fe$ during evaporation from Wang et al.[40] (Fig. 2) now allows reconstructing the $\delta^{18}O$ of the I-type spherules before

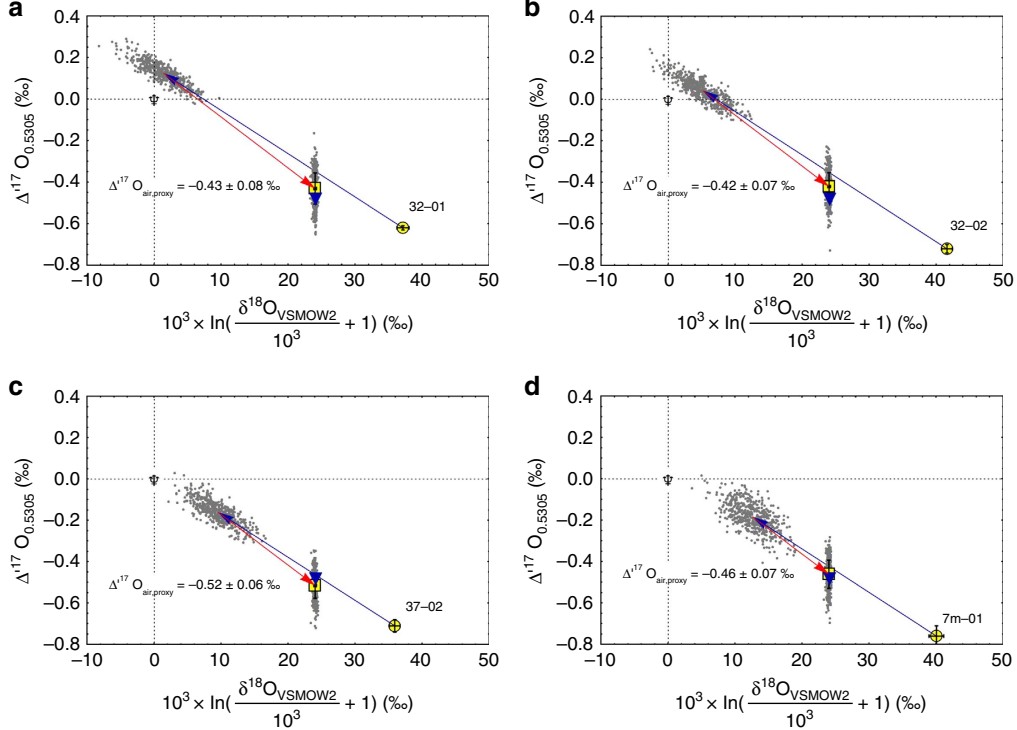

**Figure 4 | Reconstruction of the $\Delta'^{17}O$ of upper atmospheric oxygen.** Plot of $\Delta'^{17}O$ versus linearized $\delta^{18}O$ showing the result of the model calculations (including MonteCarlo error calculation ($N=500$) grey dots) for I-type spherules 32-01 (**a**), 32-02 (**b**), 37-02 (**c**) and 7 m-01 (**d**). The measured composition of the I-type spherules is displayed (yellow circle). The oxygen isotope composition is controlled by oxidation (red arrow) and evaporation (blue arrow). The resultant $\Delta'^{17}O$ (at given $\delta^{18}O=24$‰; this study) is displayed (yellow square) along with the composition of modern tropospheric $O_2$ (filled upside-down triangle). Given are $1\sigma$-error bars.

evaporation ($f=0$; Equation 2).

$$\ln\left(\frac{\delta^{18}O_{\text{before evaporation}}}{10^3}+1\right) = \ln\left(\frac{\delta^{18}O_{\text{spherule}}}{10^3}+1\right)$$
$$-1.18 \cdot \ln\left(\frac{\delta^{56}Fe_{\text{spherule}}}{10^3}+1\right) \quad (2)$$

The difference in $\delta^{18}O$ between the spherules before evaporation and air oxygen gives the degree of fractionation during the oxidation step. We obtained values between $-22$ and $-12$‰ during oxidation of the infalling Fe,Ni alloys (Fig. 3). The greater magnitudes of fractionation compared with our experiments ($-4$‰) and iron meteorite fusion crust ($\sim -9$‰; ref. 4) is attributed to the much shorter heating time of I-type spherules[10] compared with iron meteorites and experiments. Shorter heating and oxidation time prevents equilibration between oxides and air oxygen.

For the reconstruction of the $\Delta'^{17}O$ of the upper mesospheric oxygen, we need the $\theta_O$ values for the oxidation ($0.506 \pm 0.003$; Fig. 1) and evaporation. Wang et al.[40] measured $\theta_O=0.5096$ for their evaporation experiments. As in case of oxidation, the low $\theta_O$ value is indicative of kinetic fractionation during evaporation[37]. The observed $\theta_{Fe}$ values during evaporation ($\theta_{Fe}=0.671$; ref. 40) and for the I-type spherules from this study ($\theta_{Fe}=0.674$) are significantly lower than the high-$T$ equilibrium value of $\theta_{Fe}=0.687$ (see ref. 37 for details) and support kinetic fractionation.

The results of the calculation are listed in Supplementary Table 3 and illustrated in Fig. 4. The gas that oxidized the studied Antarctic I-type spherules had $-0.510 \le \Delta'^{17}O \le -0.420$‰ (mean $-0.460 \pm 0.020$‰; this study;[14,17] which is in excellent agreement with measured $\Delta'^{17}O = -0.469$‰ for the modern troposphere (Fig. 4).

Our data from oxygen and iron isotope analyses of Antarctic I-type spherules are consistent with an oxygen source with a $\Delta'^{17}O$ similar to that of modern tropo- and stratospheric molecular oxygen within $\pm 0.02$‰. No oxygen reservoir with a markedly different $\Delta'^{17}O$ participated in the oxidation of I-type spherules, suggesting that the Earth atmosphere is isotopically homogenous up to the mesosphere in $\sim 70-80$ km (Fig. 5).

Our results imply that the oxygen isotope composition ($\Delta'^{17}O$) of the bulk atmosphere can be reconstructed from combined oxygen and iron isotope data of I-type cosmic spherules. This has an important implication for the reconstruction of past atmospheric $CO_2$ levels. Blunier et al.[42] showed that the $\Delta'^{17}O$ of atmospheric molecular oxygen, indeed, varies with $CO_2$ partial pressures. This is predicted from experiments[12] and mass balance modeling[14]. The oxygen and iron isotope composition of unaltered fossil I-type cosmic spherules[4,30–33] will thus provide information on the $\Delta'^{17}O$ of the ancient atmosphere and past $CO_2$ levels. The resolution of the calculated $\Delta'^{17}O_{O_2}$ is $\sim 0.07$‰ (single I-type cosmic spherule; see Fig. 4), which translates (at modern GPP) to an uncertainty in the $CO_2$ mixing ratio of $\sim 200$ p.p.m.[14]. Lower than modern GPP levels would lead to an even higher resolution of calculated $CO_2$ levels. The reconstruction of $CO_2$ levels based on $^{17}O$ in I-type spherules is therefore considerably more precise than $CO_2$ reconstruction from $^{17}O$ of sulfate[1] and reaches far more back into Earth history than $^{17}O$ from air inclusions in ice cores[42] and fossil mammal bioapatite[2,13]. However, to use $\Delta'^{17}O$ of air oxygen as paleo-$CO_2$-barometer the GPP at that time needs to be known[14]. This may limit the usability of the new proxy. The apparent disadvantage, however, can be turned into a fortune. If the atmospheric $CO_2$ concentration is known from other, independent proxies[43], $\Delta'^{17}O$ of atmospheric molecular oxygen

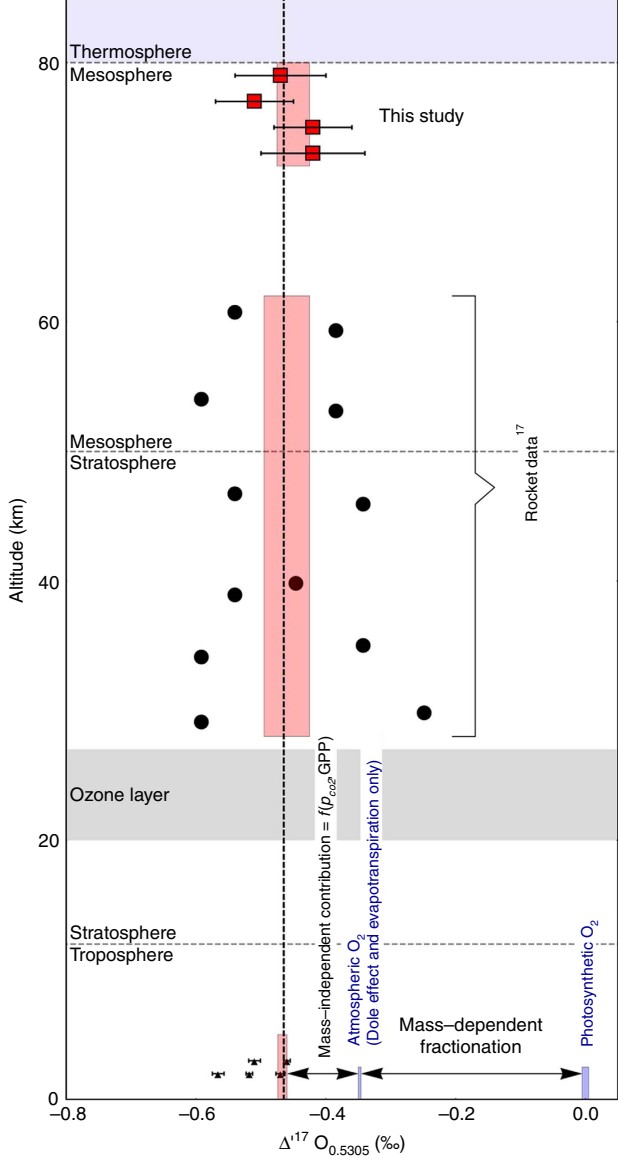

**Figure 5 | Isotope profile through the Earth atmosphere.** Diagram showing the Δ'$^{17}$O profile of the modern atmosphere. The data up to 61 km are measurements on air samples (solid circles:[17]; solid triangles:[14,18–20,54]). These data agree well with the mesospheric I-type spherule proxy data from this study (solid red squares with 1σ-error bars).

in combination with mass balance modeling[14] turns into a proxy for the GPP. There is little doubt that I-type cosmic spherules were deposited during the entire geological history of the Earth. The question is whether sufficiently large and unaltered fossil I-type spherules can actually be recovered from sediments. The recent find of unaltered 2.7 Ga old I-type cosmic spherules[33] is very promising in this respect.

## Methods

**Sampling and experiments.** We studied a total of 21 aliquots of four samples for oxygen isotopes and one aliquot of the four samples each for iron isotopes. The samples are part of the Transantarctic Mountain collection[44]. The sample sizes range between 400 and 550 μm with masses between 160 and 370 μg. The samples were inspected for weathering products by optical microscopy. No weathering products (for example, brownish ferrihydrite or goethite) were observed.

The spherule densities were determined prior to crushing. Their diameters and masses were measured and from this their densities calculated (Supplementary Table 4). The I-type spherule densities vary between 4.3 and 4.8 g cm$^{-3}$ (Supplementary Table 4).

The density data confirm that the I-type spherules are composed of Fe,Ni oxides with little or no remaining Fe,Ni metal. The studied samples fall within the density range (5.0 ± 0.5 g cm$^{-3}$) observed by Feng et al.[45]. Wüstite ([Fe,Ni]O$_{0.94}$) has a density of 5.7 and magnetite ([Fe,Ni]$_2$FeO$_4$) of 5.2 g cm$^{-3}$. In contrast, iron metal has a density of 7.9 g cm$^{-3}$. The apparent lower density of the spherules compared to wüstite and magnetite is explained by ∼20 vol.% pore space.

The samples are all spherical due to melting during their atmospheric entry. The bulk elemental composition and the mineralogy of the studied spherules were not determined. Electron microprobe analysis of I-type spherules from the same collection yielded 91 ± 5 wt.% FeO, 2.8 ± 0.5 wt.% NiO, and MgO, Al$_2$O$_3$, and <0.5 wt.% SiO$_2$ (ref. 44). This composition is similar to results of Engrand et al.[24] and Herzog et al.[28] who report values of 92–93 wt.% FeO and 4–5 wt.% NiO.

For isotope analysis, spherules were wrapped in Al foil and gently crushed in a steel mortar. We obtained 21 aliquots (4–8 per spherule) with masses of 20–50 μg. As magnetite is the dominant phase in I-type spherules[24,46], we used terrestrial magnetite along with NBS-28 quartz for tests. For NBS-28, we adopted a δ$^{18}$O = 9.65‰ and Δ'$^{17}$O = −0.054‰ (Δ'$^{17}$O from[47], with revision from[34]).

I-type spherules form by oxidation of Fe,Ni alloys at high temperatures during their atmospheric entry. We conducted three metal oxidation experiments at the University of Göttingen to study the oxygen isotope fractionation associated with high temperature oxidation of metal in air. A powdered iron base alloy with 7.5 wt.% Ni and 0.6 wt.% Co was used as an analogue material for I-type spherules. Between 0.8 and 1.1 mg powder was placed on a ceramic plate in the hot zone of a Gero HTRV vertical gas-mixing furnace. The furnace was flushed with 300 ml min$^{-1}$ air. Oxidation occurred between 1,510 and 1,590 °C for 30 min.

Air samples were taken at the Göttingen University North Campus outside the Geoscience Building (51°33′23″ N 9°56′46″ E). The air was taken from the balcony on the 4th floor using a 5 ml syringe yielding ∼1 ml standard temperature and pressure O$_2$ gas.

**Oxygen isotope analyses.** Variations in stable oxygen isotope ratios of a sample are expressed in form of the δ notation relative to the ratios in VSMOW2 water (Equation 3) with i standing for masses 17 and 18:

$$\delta^i O_{sample} = \frac{\frac{^iO}{^{16}O_{sample}}}{\frac{^iO}{^{16}O_{VSMOW2}}} - 1 \qquad (3)$$

Deviations from an otherwise close correlation between δ$^{17}$O and δ$^{18}$O are expressed in form of the Δ'$^{17}$O notation (Equation 4). We choose a reference line with slope 0.5305 and zero intercept. Deviations of Δ'$^{17}$O from zero can be caused by both, non-mass-dependent and mass-dependent processes.

$$\Delta'^{17}O^{sample} = \ln(\delta^{17}O^{sample} + 1) - 0.5305 \cdot \ln(\delta^{18}O^{sample} + 1) \qquad (4)$$

The fractionation between two reservoirs (A, B) is expressed in form of the fractionation factor α (Equation 5). The reservoirs could be two phases in equilibrium or products (B) and educts (A) of a kinetic process.

$$\alpha_{A-B}^{i/16} = \frac{\delta^i O^A + 1000}{\delta^i O^B + 1000} \qquad (5)$$

The i in Equation 5 stands for isotopes with masses 17 and 18. For mass-dependent processes, the relation between α$^{17/16}$ and α$^{18/16}$ is linked through the triple oxygen isotope fractionation exponent θ$_O$ (Equation 6).

$$\alpha_{A-B}^{17/16/O} = (\alpha_{A-B}^{18/16})^{\theta_O} \qquad (6)$$

For oxygen θ$_O$ varies between 0.5000 and 0.5305 (refs 37,48,49). Only for very small α values, θ values may fall outside the 0.5–0.5305 range[50]. Such effects are neglected here. As a rule, low θ values are associated with kinetic effects, whereas higher θ values are associated with equilibrium fractionation processes.

The triple oxygen isotope ratios of I-type spherules and the high-T oxidation experiment run products were analysed at the University of Göttingen on O$_2$ extracted by infrared laser fluorination[51], following the protocol described in Pack et al.[2]. In brief, sample O$_2$ was liberated by laser fluorination (F$_2$) and analysed in continuous flow mode in a Thermo MAT253 gas source mass spectrometer. NBS-28 quartz was used for normalisation relative to VSMOW2 scale (δ$^{17}$O = 5.04‰, δ$^{18}$O = 9.65‰, Δ'$^{17}$O = −0.054‰; using the revised calibration of San Carlos olivine from[34]). The total masses of the spheres were 160–370 μg and thus suitable for measurement of multiple aliquots of a single spherule. For the oxidation experiments, ∼0.6–1 mg aliquots were analysed.

Oxygen from air was extracted using the same line that was described by Pack et al.[34] for their water (including VSMOW2 and SLAP2) and silicate analyses (San Carlos olivine) (Fig. 6).

For each extraction, 5 ml aliquots of air standard temperature and pressure (STP) were injected through a liquid nitrogen cooled glass U-trap (for removal of moisture and CO$_2$; 'trap 7', Fig. 6). The dry, CO$_2$ free mixture of Ar, N$_2$ and O$_2$ was transferred to 'trap 2' that was filled with 5 Å molecular sieve pellets. In an early protocol (S01–S05; Supplementary Table 1), Ar was separated from O$_2$ at −100 °C, which resulted in very long trapping times. In an improved protocol (B01–BP2; Supplementary Table 1), separation of Ar was performed using the cryo unit of the Hewlett-Packard 5890 gas chromatograph at −80 °C. After Ar had passed through the gas chromatograph (monitored using a Pfeiffer Prisma

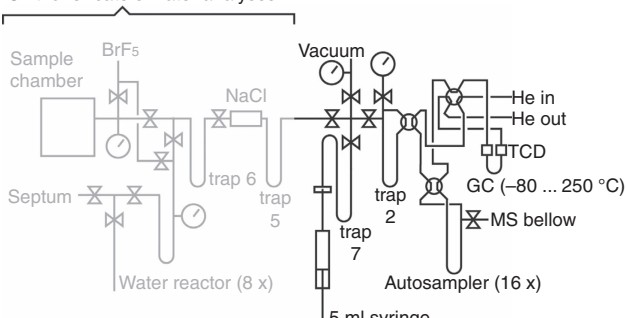

**Figure 6 | Illustration of the extraction line.** Sketch of the extraction line used for the air measurements. The same line was used for the measurement of VSMOW2, SLAP2 and San Carlos olivine[34].

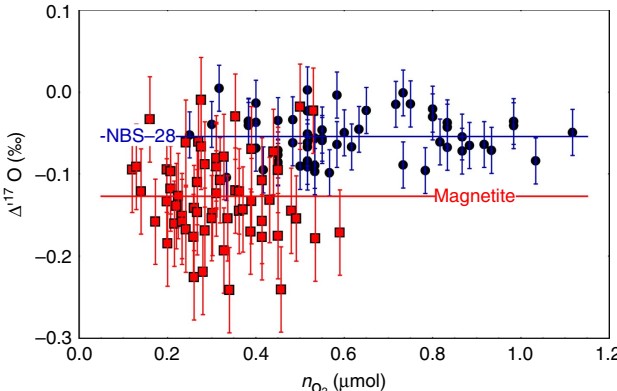

**Figure 7 | Results of oxygen isotope analyses of small samples.** Plot of $\Delta'^{17}O$ of NBS-28 quartz (solid blue circles) and magnetite (solid red squares) versus amount of $O_2$ (in moles) in the samples. The continuous flow measurements were normalized to NBS-28 with $\Delta'^{17}O = -0.054‰$. The smallest magnetite samples had masses of only $\sim 10\,\mu g$. No systematic variation in $\Delta'^{17}O$ is observed with sample size. Errors are $1\sigma$ standard deviations.

quadrupole mass spectrometer at the end of the He capillary; 'He out' in Fig. 6), temperature was raised to $-30\,°C$ for the elution of $O_2$ and separation of $N_2$. The improved protocol is similar to the protocol described by Young et al.[14]. The purified $O_2$ was analysed for $\sim 60-90\,min$ in dual inlet mode.

To test accuracy and precision of laser $F_2$ in combination with continuous flow mass spectrometry of small samples, a set of experiments were performed on magnetite (as analogues of I-type spherules) and NBS-28 quartz (Fig. 7).

Our tests on NBS-28 quartz and magnetite (Fig. 7) show that precise analyses of $\delta^{18}O$ and $\Delta'^{17}O$ are possible down to 10 µg samples. The blank intensities corresponded to $\leq \sim 0.005\,\mu mol\,O_2$ ($\leq \sim 6\%$ of the sample). The tests reveal uncertainties in $\delta^{18}O$ of $\pm 0.6‰$ for quartz and $\pm 1‰$ for magnetite. The uncertainties in $\Delta'^{17}O$ were $\pm 0.03‰$ for quartz and $\pm 0.05‰$ for magnetite. The higher uncertainty observed for magnetite may be due to heterogeneity on small scale (for example,[52]). For our I-type spherule analyses, we adopt analytical uncertainties of 1‰ for $\delta^{18}O$ and 0.06‰ for $\Delta'^{17}O$. We have not adopted the VSMOW2-SLAP2 scaling for the continuous flow measurements since conditions during the measurement are less controlled (for example, variable peak heights and widths) and VSMOW2-SLAP2 correction for $\Delta'^{17}O$(ref. 34) would be smaller than the uncertainty in $\Delta'^{17}O$. The agreement between the spherule $\Delta'^{17}O$ from this study and conventional $F_2$ data from Clayton et al.[4] confirm that the continuous flow $\delta^{17}O$ data are on VSMOW2 scale.

**Iron isotopes.** Three iron isotopes have been analysed ($^{54}Fe$, $^{56}Fe$, $^{57}Fe$). Isotope ratios are expressed relative to the IRMM-014 standard in form of the $\delta$ notation with:

$$\delta^i Fe = \frac{\frac{^i Fe}{^{54}Fe}_{sample}}{\frac{^i Fe}{^{54}Fe}_{IRMM-14}} - 1 \tag{7}$$

and $i$ corresponding to masses 56 and 57. All $\delta^{56}Fe$ and $\delta^{57}Fe$ data are reported in ‰ variations. Variations in the three iron isotopes are coupled through the triple iron isotope exponent $\theta_{Fe}$ (Equation 8).

$$\alpha_{A-B}^{56/54} = (\alpha_{A-B}^{56/54})^{\theta_{Fe}} \tag{8}$$

The symbols A and B in Equation 8 can either stand for two phases that are in equilibrium or for educts (B) and products (A) of a reaction with associated kinetic fractionation. The high-$T$ approximation for equilibrium iron isotope fractionation is $\theta_{Fe} = 0.6784$ (ref. 37). As in case of oxygen, variations in $\theta_{Fe}$ provide insights into the fractionation process.

The iron isotope compositions were measured at the University of Bonn using a Thermo Scientific Neptune MC-ICP-MS instrument and glassware for sample introduction. Samples were measured 2–3 times, non-consecutively, during long analytical sessions of around 8 hours. Analyses were carried out in high-resolution mode with sufficient transmission to allow routine analyses of an 1 p.p.m. iron sample solutions. Each sample measurement was bracketed by two analyses of an IRMM-014 iron solution that was made up to closely match the iron concentration of the sample. The external reproducibility of the data was typically $\pm 0.07‰$, whereas the internal reproducibility was generally about a factor of two better. A detailed description on sample preparation and mass spectrometry can be found in Hezel et al.[53] and Hezel et al.[36].

**Monte Carlo simulation.** The errors in $\delta^{18}O$ and $\Delta'^{17}O$ of the upper atmospheric oxygen were estimated using a Monte Carlo approach. The composition was computed 500 times. Each input parameter was varied independently for each run (Supplementary Table 5). We used a normal distribution for the variation within the respective error interval. The computation was performed using the Mathematica software.

**Data availability.** The authors declare that the data supporting the findings of this study are available within the paper and its supplementary information.

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

## Acknowledgements

The authors thank Reinhold Przybilla and Ingrid Reuber for assistance in the laboratory. The collection and characterisation of the micrometeorites studied in this work was supported by the Italian MIUR (grant PNRA16_00029 and PRIN2015_20158W4JZ7). The project was financially supported by Deutsche Forschungsgemeinschaft (grant PA909/11-1).

## Author contributions

A.P. conceived and supervised the project. A.H., D.C.H., M.T.S., A.-K.B., S.T.M.P., S.S. and D.H. designed the experiments, prepared the samples and conducted the analyses. L.F. provided the samples. A.P. wrote the manuscript.

## Additional information

**Competing interests:** The authors declare no competing financial interests.

