## [Peer Review File · Nature Communications]

Reviewers' comments:

Reviewer #1 (Remarks to the Author):

This is an excellent and very important paper that establishes the oxygen isotope composition of the upper atmosphere at high resolution. It clearly deserves publication in Nature Comms. There are some minor issues with English, such as the consistent use of "air O₂" and the odd strange wording "a share of" rather than "a proportion of". My field of expertise is in micrometeorites rather than in oxygen isotopes, so my review assumes that the measurements and equations used are robust. I am sure other reviewers will comment further on these aspects.

I have some general comments which the authors should address:

(1) Terrestrial alteration

These particles have terrestrial ages up to 2 Ma and may have experienced terrestrial alteration on the surface, in particular the replacement of metal by goethite/ferrihydrite. This phase contains oxygen obtained at the surface and can potentially skew the results. The authors mention that there is up to 20 vol% void space on the basis of density measurements, if this was ferrihydrite, with oxygen having a tropospheric oxygen isotope composition, could this alter the result and by how much? You might want to add a paragraph showing this isn't the case. The magnetite and wustite in spherules is resistant to terrestrial weathering and I am confident these will continue to record oxygen isotope composition at the altitude of deceleration.

(2) Altitude of Oxidation

The altitude at which oxidation occurs is crucial, this is after all an attempt to measure the oxygen isotope composition of the thermosphere. The bulk of oxygen acquired by I-types is accreted at peak deceleration. Genge, 2016 shows that for spherules 400-550 µm in diameter this is ~70 to 77 km in altitude (assuming an entry velocity of 12 km/s and entry angles >20 degrees). According to Wikipedia (that well known font of all knowledge ;)) the thermosphere is above 85 km. These measurements would appear, therefore, to be dominated by oxygen from the mesosphere (although there will still be a small component from higher altitudes). I'd recommend changing the title somewhat to "upper atmosphere". I think it is really important to discuss the likely altitudes at which most of the oxygen is accreted to the spherules.

(3) Implications

There are some statements about measuring past oxygen isotope compositions, which are absolutely true and the major importance of this work. Although there is a nice discussion of the role of global oxygen budget I think the authors could include a little more in the way of implications. I will give an example for free....Genge 2016 shows that altitude of peak deceleration is highly dependent on particle size (albeit with some variation due to entry angle). Given increased analytical resolution I-types could potentially be used to target different altitudes in the mesosphere and the thermosphere. If a means of identifying low entry angle particles could be identified on mineralogical or textural grounds, then better height resolution can be obtained.

(4) Conclusions

Presently the paper ends rather abruptly. I think it is worthy of a concluding paragraph that reinforces the paper's significance.

Review by
Matt Genge

Reviewer #3 (Remarks to the Author):

Overall Evaluation

The approach of using metal entering the atmosphere as a monitor of upper atmosphere oxygen isotope ratios is sound and well thought out. The effects of the relevant processes are presented clearly, and the use of Fe isotopes as a marker for the degree of evaporation is sound. The analytical procedures are adequate for the claims made in the paper. Overall the authors have demonstrated that oxidized metal extraterrestrial spherules are viable as records of atmospheric oxygen isotope ratios.

Significance

I am not certain as to the important implications of the results. The authors conclude their manuscript by stating that fossil I-type oxidized metal spherules could be used as monitors of past atmospheric $\delta^{17}\text{O}$ values. These results in turn can be used to constrain gross primary production (GPP) or partial pressures of CO_2 , although it must be recognized that these two values are linked and so there is a degeneracy in the interpretation (describe below). Less clear is whether the method described in this paper is superior to other methods that have been used in this arena. Air oxygen trapped in glacial ice (e.g., Blunier et al., 2002), for example, has been used to obtain ancient $\delta^{17}\text{O}$ values with implications for pCO_2 and/or GPP, and $\delta^{17}\text{O}$ values in sulfate minerals have been used to obtain estimates of pCO_2 in deep time (e.g., Bao et al. 2008). The Blunier study reports a Last-Glacial-Maximum $\delta^{17}\text{O}$ value higher than today by 0.04 per mil, which is well within the resolution of 0.07 per mil cited by the authors (line 185). There is also the question of whether a precision of about ± 100 to ± 200 ppm is useful in deep time. Firstly, it seems the cited precision in paleo- pCO_2 of ± 110 ppm is slightly too optimistic in view of recent calculations. Figure 7 from Young et al. (2014, GCA 135) suggests a resolution of more like ± 150 ppm at 100% of present-day GPP. More importantly, past applications of $\delta^{17}\text{O}$ in deep time (e.g., Neoproterozoic cap carbonates) find differences of tens of thousands of permil, making the potential increase in precision from fossil spherules of limited utility. Also, in order to advocate for using the spherules rather than ice cores or sulfates, the authors should probably address the frequency with which these spherules are found in strata as a function of geological age.

For these reasons, I submit that the study is creative and well executed, though future applications of the method are not obviously superior to existing methods.

Detailed Comments

Line 51: 57% of the negative $\delta^{17}\text{O}$ effect in the troposphere is due to the Dole effect, 33% is due to stratosphere photochemistry, and 10% results from evapotranspiration. These estimates are based on models that fit the data (Young et al. 2014, GCA 135). This does not mean that it is not a paleo- CO_2 barometer, but it is more complicated than that simple function implies, depending also on gross primary production (see fig.7 in Young et al. 2014).

Line 155: There is of course an element of circularity to the extent that the $\delta^{17}\text{O}$ obtained by this method applies to an assumed $\delta^{18}\text{O}$ value that coincides with tropospheric O_2 . I realize the point is to constrain $\delta^{17}\text{O}$, but it is of course at least possible that there might be an offset in $\delta^{18}\text{O}$ between the troposphere and atmospheric elevations of ~ 100 km. Nonetheless, it would be coincidental that the $\delta^{17}\text{O}$ happens to match the tropospheric value at the corresponding $\delta^{18}\text{O}$ if indeed the upper atmosphere were different in $\delta^{18}\text{O}$.

Line 179: Here the authors may wish to compare their results to previous predictions. Young et al. (2014) show results for a model in Table 3 suggesting a small difference between $\delta^{17}\text{O}$ of stratospheric O_2 and tropospheric O_2 of about 40 ppm. The stratosphere value is very susceptible to small differences in photochemistry kinetics and so this difference may not be real, but it illustrates that up to about 20 to 30 km no difference is expected. I don't know what the

expectations have been up to > 80 km.

Figure 4: We need a bit more information about the Monte Carlo simulations - how were the ranges chosen, were values for D_{17O} and or the fractionation exponents selected independently of the d_{18O} or α values for oxidation, for example? The Monte Carlo simulations are mentioned only in the Figure caption.

Reply to comments on manuscript *Tracing the oxygen isotope composition of the Earth's upper atmosphere using cosmic spherules*

Response to Referees

We appreciate the very helpful comments by both reviewers and have addressed the concerns.

Reviewer #1 (Dr. Matthew Genge)

[...] There are some minor issues with english, such as the consistent use of "air O₂" and the odd strange wording "a share of" rather than "a proportion of". [...]

We have corrected the manuscript and replaced "air O₂" by air oxygen. Only where it is relevant that we are talking about "O₂", we replaced "air O₂" by "molecular air oxygen". We are in line with the reviewer and have replaced "share of" by "portion of". We have also corrected a number of small typos throughout the manuscript.

(1) Terrestrial alteration

These particles have terrestrial ages up to 2 Ma and may have experienced terrestrial alteration on the surface, in particular the replacement of metal by goethite/ferrhydrite. This phase contains oxygen obtained at the surface and can potentially skew the results. The authors mention that there is up to 20 vol% void space on the basis of density measurements, if this was ferrhydrite, with oxygen having a tropospheric oxygen isotope composition, could this alter the result and by how much? You might want to add a paragraph showing this isn't the case. The magnetite and wustite in spherules is resistant to terrestrial weathering and I am confident these will continue to record oxygen isotope composition at the altitude of deceleration.

We agree with the reviewer that the presence of terrestrial oxide alteration products would, if present, have to be considered here. The selected micrometeorites were inspected by microscopy for such oxidation products and were found to be free of them.

As mentioned by the reviewer, magnetite and wüstite (the major I-type micrometeorite constituents) are very resistant to weathering; namely when stored most of the time frozen in ice. Also, calculations and observations suggest that the likelihood of preservation of metal I-type spherules decreases with size with little or no metal surviving in spherules > 150 µm (Genge, 2016); our spherules were between 400 and 550 µm. Metal is much more prone to terrestrial weathering than magnetite and wüstite. Hence, the absence of weathering products (from metal oxidation) is actually expected for our micrometeorites.

We added a short phrase clarifying the the studied samples were free of weathering products (Materials & Methods section).

(2) Altitude of Oxidation

The altitude at which oxidation occurs is crucial, this is after all an attempt to measure the oxygen isotope composition of the thermosphere. The bulk of oxygen acquired by I-types is accreted at peak deceleration. Genge, 2016 shows that for spherules 400-550 µm in diameter this is ~70 to 77 km in altitude (assuming an entry velocity of 12 km/s and entry angles >20 degrees). According to Wikipedia (that well known font of all knowledge ;)) the thermosphere is above 85 km. These measurements would appear, therefore, to be dominated by oxygen from the mesosphere (although there will still be a small component from higher altitudes). I'd recommend changing the title somewhat to "upper atmosphere". I think it is really important to discuss the likely altitudes). I'd recommend changing the title somewhat to "upper atmosphere". I think it is really important to discuss the likely altitudes at which most of the oxygen is accreted to the spherules.

We agree with the reviewer here. Some oxygen likely sources from the thermosphere, but, as stated by the reviewer, most is likely taken up in the mesosphere. Accordingly, we have changed to "mesosphere" throughout the manuscript and in Fig. 5 (here, an altitude between 75 and 80 km is assumed; Genge, 2016). We also changed the x-scale from ppm to permil (‰) to keep the units consistent throughout the text.

(3) Implications

There are some statements about measuring past oxygen isotope compositions, which are absolutely true and the major importance of this work. Although there is a nice discussion of the role of global oxygen budget I think the authors could include a little more in the way of implications. I will give an example for free....Genge 2016 shows that altitude of peak deceleration is highly dependent on particle size (albeit with some variation due to entry angle). Given increased analytical resolution I-types could potentially be used to target different altitudes in the mesosphere and the thermosphere. If a means of identifying low entry angle particles could be identified on mineralogical or textural grounds, then better height resolution can be obtained.

Our data show that the Earth atmosphere is isotopically homogenous in oxygen up to ~ 80 km. This means that (in combination with Fe isotope analyses), triple oxygen isotopes of I-type cosmic spherules can be used to determine the $\Delta^{17}\text{O}$ of the bulk atmospheric molecular oxygen. The most important implication is that fossil I-type micrometeorites can be used as high-resolution (compared to other approaches) paleo-CO₂-proxy.

Surely, combining entrance modeling and isotope data would provide additional information on the exact entry history. This is, however, beyond the scope of this paper.

(4) Conclusions

Presently the paper ends rather abruptly. I think it is worthy of a concluding paragraph that reinforces the papers significance.

We modified the last paragraph of the manuscript accordingly.

Reviewer #3 (anonymous)

[...] I am not certain as to the important implications of the results. The authors conclude their manuscript by stating that fossil I-type oxidized metal spherules could be used as monitors of past atmospheric $\delta^{17}\text{O}$ values. These results in turn can be used to constrain gross primary production (GPP) or partial pressures of CO_2 , although it must be recognized that these two values are linked and so there is a degeneracy in the interpretation (describe below). Less clear is whether the method described in this paper is superior to other methods that have been used in this arena. Air oxygen trapped in glacial ice (e.g., Blunier et al., 2002), for example, has been used to obtain ancient $\delta^{17}\text{O}$ values with implications for pCO_2 and/or GPP, and $\delta^{17}\text{O}$ values in sulfate minerals have been used to obtain estimates of pCO_2 in deep time (e.g., Bao et al. 2008). The Blunier study reports a Last-Glacial-Maximum $\delta^{17}\text{O}$ value higher than today by 0.04 per mil, which is well within the resolution of 0.07 per mil cited by the authors (line 185). [...]

We now have better clarified in the last paragraph of the manuscript that atmospheric CO_2 and global primary production are closely linked (this was, however, already explained in the introduction part).

The approach described here for the first time is a) much more precise (by at least an order of magnitude!) than the sulfate approach by Bao et al. (2008) and b) reaches back beyond the 800,000 years covered by ice cores (Blunier et al., 2002). Also, for ice cores, one would not use $\Delta^{17}\text{O}$ as CO_2 proxy but measure the CO_2 directly from air bubbles. The approach described here has great potential for reconstructing pCO_2/GPP throughout the entire Earth history. It is not the aim to resolve small < 100 ppm variations in atmospheric CO_2 , e.g. between glacials and interglacials. Instead, the I-type spherules shall be used to resolve > 100 ppm variations in atmospheric CO_2 throughout Earth history.

As was recently shown by Tomkins et al. (2016), I-type cosmic spherules as old as 2.7 Ga have been well-preserved in carbonates. Some even contain metal. We have stressed out the potential of our new approach in the last part of the manuscript.

[...] Firstly, it seems the cited precision in paleo- pCO_2 of ± 110 ppm is slightly too optimistic in view of recent calculations. Figure 7 from Young et al. (2014, GCA 135) suggests a resolution of more like ± 150 ppm at 100% of present-day GPP. More importantly, past applications of $\delta^{17}\text{O}$ in deep time (e.g., Neoproterozoic cap carbonates) find differences of tens of thousands of permil, making the potential increase in precision from fossil spherules of limited utility. Also, in order to advocate for using the spherules rather than ice cores or sulfates, the authors should probably address the frequency with which these spherules are found in strata as a function of geological age [...]

We have re-assessed the uncertainty of our approach. Given an uncertainty in reconstruction of atmospheric

O₂ 0.07 ‰, the uncertainty in CO₂ mixing ratio is 180 ppm. We now give a rounded estimate of the uncertainty of 200 ppm instead of 100 ppm as in the original version. The precision of the approach increases with decreasing GPP.

We disagree with the reviewer that the increased precision is of limited use. For sulfate from the Neoproterozoic cap carbonates, Bao et al. (2008) reported a CO₂ mixing ratio of 12,000 ppm (with an error of + 15,000 and -6000 ppm). We believe that there's surely need for a better proxy here. We do not agree with the reviewer that more precise data are of little use. Instead, the Bao et al. (2008) study clearly illustrates that an improvement is highly desired here.

There are no data available yet on the frequency of cosmic spherules in sediments. It is now clarified in the text (last part of manuscript) that a) there is no reason to assume that I-type spherules were deposited throughout the entire Earth history and that b) they will thus provide an excellent record on the composition of the atmosphere and GPP. There is no systematic search yet been undertaken on the abundance of these objects in sediments. The recent find of unaltered 2.7 Ga (!) old cosmic spherules, however, is highly encouraging. This has also been outlined in the text now.

[...] Line 51: 57% of the negative D¹⁷O effect in the troposphere is due to the Dole effect, 33% is due to stratosphere photochemistry, and 10% results from evapotranspiration. These estimates are based on models that fit the data (Young et al. 2014, GCA 135). This does not mean that it is not a paleo-CO₂ barometer, but it is more complicated than that simple function implies, depending also on gross primary production (see fig.7 in Young et al. 2014). [...]

We agree with the reviewer and refer to the detailed mass balance model published by Young et al. (2014) as a better basis for the interpretation of the data as previously cited Bao et al. (2008) work. We also point out that three factors affect the Δ¹⁷O of air O₂: Dole-effect, evapotranspiration, and mass-independent fractionation in the stratosphere.

[...] Line 155: There is of course an element of circularity to the extent that the D¹⁷O obtained by this method applies to an assumed d¹⁸O value that coincides with tropospheric O₂. I realize the point is to constrain D¹⁷O, but it is of course at least possible that there might be an offset in d¹⁸O between the troposphere and atmospheric elevations of ~ 100 km. Nonetheless, it would be coincidental that the D¹⁷O happens to match the tropospheric value at the corresponding d¹⁸O if indeed the upper atmosphere were different in d¹⁸O. [...]

There is no observed shift in δ¹⁸O and Δ¹⁷O between the troposphere and the stratosphere up to 61 km (Thiemens et al., 1996). Also, the model by Young et al. (2014) does not suggest any stratospheric shift in δ¹⁸O. We have now corrected the altitude estimate to 70 – 80 km (see reviewer #1). Based on observation (Thiemens et al., 1995) and model (Young et al., 2014) we assume that our δ¹⁸O estimate is robust. A small shift in δ¹⁸O, however, would not have much effect (outside error limit) on the resultant Δ¹⁷O.

[...] Line 179: Here the authors may wish to compare their results to previous predictions. Young et al. (2014) show results for a model in Table 3 suggesting a small difference between D¹⁷O of stratospheric O₂ and tropospheric O₂ of about 40 ppm. The stratosphere value is very susceptible to small differences in photochemistry kinetics and so this difference may not be real, but it illustrates that up to about 20 to 30 km no difference is expected. I don't know what the expectations have been up to > 80 km. [...]

The reviewer refers to table 3 in Young et al. (2014). The table reports the following data:

Troposphere (O₂): $\delta^{17}\text{O} = 11.887 \text{ ‰}$, $\delta^{18}\text{O} = 23.212 \text{ ‰}$ → $\Delta^{17}\text{O} = -0.427 \text{ ‰}$

Stratosphere (O₂): $\delta^{17}\text{O} = 11.886 \text{ ‰}$, $\delta^{18}\text{O} = 23.212 \text{ ‰}$ → $\Delta^{17}\text{O} = -0.428 \text{ ‰}$

giving a model difference in $\Delta^{17}\text{O}$ between the tropo- and stratosphere of only 1 ppm, not, as stated, 40 ppm. Because a 1 ppm difference is so much smaller than the uncertainty of our reconstruction from cosmic spherules, we therefore did not change the manuscript here.

[...] Figure 4: We need a bit more information about the Monte Carlo simulations - how were the ranges chosen, were values for $\Delta^{17}\text{O}$ and or the fractionation exponents selected independently of the $\delta^{18}\text{O}$ or alpha values for oxidation, for example? The Monte Carlo simulations are mentioned only in the Figure caption. [...]

We agree with the reviewer and gave a short outline of the method applied in the Materials & Methods section. We also added an error envelope to the $\delta^{18}\text{O} - \delta^{56}\text{Fe}$ relation in Figure 2.

We appreciate the reviewer's comments and hope that the manuscript is now suitable for publication,

REVIEWERS' COMMENTS:

Reviewer #1 (Remarks to the Author):

I think the authors have dealt very well with all reviewers comments. I must admit I agree with the authors comments to reviewer 2 in regard to the significance of the work. Assessments of atmospheric oxygen isotope obtained on cosmic spherules is a significant advance since it allows measurement today and in the past at altitude. Reviewers suggestion that surface measurements are sufficient (air in ice or sulphates) are clearly not valid given that they later discuss the uncertainty on isotope compositions at altitude. I recommend publication.